# Climate Competencies of Finnish Gifted and Average-Ability High School Students

**Sakari Tolppanen** [1,*] **, Jingoo Kang** [1] **and Kirsi Tirri** [2]

1    School of Applied Educational Science and Teacher Education, University of Eastern Finland, 80110 Joensuu, Finland; jingoo.kang@uef.fi
2    Faculty of Educational Sciences, University of Helsinki, 00170 Helsinki, Finland; kirsi.tirri@helsinki.fi
*    Correspondence: sakari.tolppanen@uef.fi

**Abstract:** In the face of global issues such as climate change, the world needs action competent, transformationally gifted citizens, who are willing to step up and take responsibility for a better future. However, empirical evidence on what supports the development of transformational giftedness is limited. Furthermore, the relationship between academic giftedness and transformational giftedness has not been clearly pronounced. The purpose of this study is to address this research gap by examining students' climate competencies. A total of 1703 students from five Finnish high schools (grades 10–12) participated in this study. Using a questionnaire, students' climate change knowledge, values, willingness to take action, sense of responsibility, environmental concern, and perceptions on how climate change issues are dealt with in school were examined. Four of the schools were general education high schools, while one was for students formally identified as gifted students. The findings indicate that academically gifted students in both general education schools and the gifted school show more climate competencies than average-ability students. Furthermore, gifted students that attended the school for gifted students show more climate competencies than the gifted students from general education schools. Based on the findings, the paper discusses how the development of transformational giftedness can be better supported in education.

**Keywords:** climate change education; academic giftedness; transformational giftedness; transformational education; action competence

## 1. Introduction

Climate change is one of the most pressing global issues of our time, with potentially catastrophic consequences for societies and ecosystems. In order to mitigate and adapt to climate change, the UN has acknowledged that everyone needs to do their part: governments, businesses and individuals [1]. In this transition, education has much to contribute. In recent years there has been a realization that climate education requires increased emphasis on supporting the development of students' competencies, values, attitudes and helping them gain more rigorous knowledge about the complex relationships between humans and their habitats, as well as the rebalancing of priorities and commitments that are involved in striving for sustainability. For instance, UNESCO ([2], p. 36) has stated that schools should encourage students to "re-evaluate [their] worldview and everyday behaviours", in light of what is necessary for climate change mitigation. In practice, such a transition requires students to develop green competencies, meaning that they are capable of systems thinking and future thinking, show awareness towards sustainability challenges, including ethical and social justice dimensions, are capable of examining their underlying values and show agency to participate in impactful action, both collectively and individually, now and in the future [3]. In essence, education needs to be transformative, meaning that education develops students to become autonomous, critical thinkers, and supports them in examining their conceptual foundations, helping them make changes to their frame of reference, if necessary [4].

The ideal, is that through the process of transformational education, students become action competent, meaning that they become active citizens in a democratic society, taking both direct action and indirect action. More recently, Sternberg [5] has coined the term transformational giftedness to describe a similar thing as action competence but bringing the focus to exceptional individuals. Namely, transformational giftedness means that students have exceptional ability or talent that enables them to make one or more extraordinary and meaningful contribution that helps make the world a better place [5]. According to Sternberg et al. [6], such transformation can happen on two levels. First, it can be Self-transformational, where a "positive, meaningful and possibly enduring difference" happens within oneself. This is often the preliminary to the second type of transformational giftedness, called other-transformational, where one aims to make a positive enduring difference to the world. Though action competence and transformational giftedness resemble each other in many regards, in this study we use the term transformational giftedness, as it better describes the focus of this study, which is to examine the climate competencies of academically gifted students.

While helping students develop transformational giftedness is something to strive for, this poses enormous challenges for formal education. First, climate change education is not yet strongly present in the formal curricula of many countries, the curricula often continue to focus on the causes of climate change rather than needed actions and behavioral changes, and there continues to be limited room for discussions on values and ethics [7]. That said, some countries have more possibilities to implement climate change education than others. In Finland, where this study takes place, sustainability issues have been included as one of the four core values of the curriculum in secondary school since 2016 (see [7,8]). This means that sustainability issues and climate issues can and should be implemented into all school subjects. However, as teachers in Finland are given a lot of autonomy, the subject-specific curriculum does not give clear guidelines on how sustainability and climate issues should be implemented into education. In practice, this means that climate change education tends to focus on knowledge creation, and it is very much up to an individual teacher how they implement climate change education in practice [9]. Despite the variance in CC-Ed implementation, the curriculum provides ample opportunities for teachers in Finland to help develop students' general competencies (see [10]). Authors [11] have argued that the concept of transformational giftedness adheres very well to the educational philosophy, the German Bildung tradition, on which education in Finland and in the Nordic countries is based. This philosophy aims at educating individuals to become competent citizens who actualize their individual talents and benefit society with their competences. In Finland academic achievement is not seen as the only aim of schooling but development of the whole person including moral reasoning and behavior are also emphasized [8,11]. Therefore, the Finnish curriculum may not present as many barriers to provide transformative education as the curriculum of some other countries.

Second, we don't yet fully understand what results in action competence or transformative education (see [12]). A relatively recent literature review does give some guidelines, as it highlights that impactful climate change education should be personally relevant, engage students, foster deliberative discussion, provide interactions with scientists, address misconceptions and implement community projects [13]. However, most of the studies in the review focus on the educational impact of climate knowledge, so further studies are needed to examine what type of education is transformational from a more holistic perspective, meaning it has long-lasting impacts on students' attitudes, values and willingness to take personal and societal action.

Third, education that is transformative for one student, may not be so for all. For instance, a recent study found that a university course on holistic climate change education was more transformational for non-STEM and female students than for others [14]. Therefore, further research is needed on how to make education transformational for all, or at least for most students.

What makes transformational education especially hard, is that it is not only about gaining more knowledge [15]. In fact, studies have shown that individuals may be reluctant to change their views even when presented with compelling evidence which is not in line with their views (see, [16,17]). Therefore, many other factors, such as attitudes (see, e.g., [18]), worldview [19], ease of taking action [20] and values [21,22] are at play. Individuals are also affected by their biases, such as self-bias and intragroup bias. This translates to individuals preferring to perform low-impact actions themselves, while expecting others to do high-impact actions (e.g., [5]). Additionally, responsibility is often deflected onto governments and businesses, as individuals see their own role in mitigating climate change as limited [23,24]. In addition, psychological and social factors, including perceived behavioral control, moral obligations, societal expectations and norms effect individuals' willingness to take action (see, e.g., [18,25,26]). In practice, this also means that for a student to become transformationally gifted in climate change issues, they need not only good cognitive skills, but also an interest in moral and ethical issues, the willingness to take moral responsibility, social support and the discipline and willpower to overcome both psychological and social barriers. As this requires a lot from an individual, this study seeks to increase our understanding on what factors help develop transformational giftedness in students. More specifically, we seek to understand how academic giftedness and the school environment may enhance transformational giftedness by helping them develop competencies. Despite the extensive literature on factors effecting climate action, there is a research gap in examining the effect of academic achievement on students' climate competencies. This study aims to address that research gap.

*Academically Gifted Students*

Gagné's [27] differentiated model of giftedness and talent 2.0 (DMGT 2.0) is a comprehensive framework to understand the development of gifts into talents in different domains. According to the model, the gifts can be developed into talents in the areas of science and technology, arts, sports, and athletics. Talent development is a process that involves systematic effort from an individual with a significant amount of time and other resources and a structured educational program. Gagné sees giftedness as potential that can be developed further with appropriate levels of intrapersonal and environmental factors. He also defines a gifted individual as one among the top 10% of age peers in at least one ability domain. In line with Gagné's definition for giftedness, in this study we define academically gifted students as those whose final grade from secondary school (i.e., grade 9) was among the top 10% of the participants. Previous studies show that there is a positive relationship between academic achievement and environmental awareness (e.g., [28]) and that academically gifted students rank higher in moral reasoning and ethical sensitivity than their average-ability peers [29–31]. Naturally, academic achievement also coincides with more knowledge on a given school topic. Furthermore, gifted students have been characterized as having a high sense of responsibility, as well as a keen interest in working with issues that involve their lives and global issues [32,33]. Gifted students also tend to be good problem-solvers, enjoying tackling big challenges [30,34]. As these characteristics describe climate competent citizens, and many are essential to becoming transformationally gifted, we hypothesize that academically gifted students may show more readiness towards transformational giftedness than their peers. In other words, we view climate competence as a prerequisite for transformational giftedness. However, as Sternberg et al., discuss, academic giftedness will not automatically result in transformational giftedness [6]. Rather, transformational giftedness needs to be nurtured through education and social interactions. Therefore, our second hypothesis is that academically gifted students attending a school for gifted students may show more readiness towards transformational giftedness than gifted students in general education schools. Accordingly, this study aims to explore the following two research questions:

1. How do academically gifted students' climate competencies differ from average-ability students' competencies?
2. What type of effect does a school have on academically gifted students' climate competencies?

## 2. Materials and Methods

### 2.1. Sample and Data Description

The data for this study was collected in the fall of 2021 from five Finnish high schools, located in different parts of the country (Helsinki, Vantaa, Tampere, Mikkeli and Kajaani). Four of the selected schools were general education schools, called Normal Schools in this study. These schools required students to have completed secondary school with moderate to good final average grades. The four Normal Schools can also be considered representative of a typical Finnish high school, as the mean score of the participants was not much higher than the national average of high school students (8.83 vs. 8.67) (see [35]). The fifth school participating in this study was a more homogenous school of gifted students. To get accepted to this school, students must have an excellent final average grade from secondary school (median 9.71).

The research was conducted following the guidelines of the Finnish Advisory Board on Research Integrity TENK [36]. Following the guidelines, ethics approval was not required for the study. Approval for the study was given by the municipalities or school principles. Furthermore, the caregivers of the students were informed about the study in advance, giving them the opportunity to decide, together with their child, whether to participate in the study or not. Though all the students were encouraged to join the study by filling out a questionnaire during class time, it was clearly stated to them, both before data collection and in the online form, that participation was voluntary. Furthermore, students were informed that they can withdraw from the study at any point, even after the completion of data collection.

Out of the 2970 students attending the schools, 2191 completed the questionnaire. After omitting participants who incorrectly answered the two control questions, the remaining sample size was 1973 students. The Finnish National Agency of Education (FNAE) was contacted to receive information regarding the participating students' final grades from secondary school. After omitting students whose secondary school grades could not be tracked, the final sample size was 1703 participants. Out of these students, 670 were at the beginning of grade 10, 614 were in grade 11 and 419 in grade 12. Further information on the schools is provided in Table 1. As noted in the table, the schools have different emphases. In practice, this means that schools provide more of certain courses, giving students the *opportunity* to delve deeper into some subjects. This also means that students' interests may determine which schools they are attracted to. As the school for gifted students has a science focus, we examined the course descriptions of their extra courses. Based on the descriptions there is no reason to believe that students in that school are exposed more to climate change and sustainability issues in their science classes than students in other schools. Furthermore, as climate change is a multidisciplinary issue, not only an issue to be addressed in science class, we cannot assume that merely having a science focus would mean that students are exposed more to climate change issues than in another schools. Unfortunately, it was beyond the scope of this study to conduct interviews and classroom observations to determine what really happens in class.

### 2.2. Measures

The initial questionnaire consisted of 11 sections and 97 questions and took around 30–40 min for students to complete. Among them, we used 47 items that were relevant to this study aim as presented in Table 2 (See Appendix A for list of questions used). Reliability and validity of the measurement have been reported in the following section.

**Table 1.** Background information of the five schools that took part in the study.

| | School 1 | School 2 | School 3 | School 4 | School 5 |
|---|---|---|---|---|---|
| School type | Normal Schools (Public school for all students) | | | | Gifted School (Public school for gifted students) |
| Location | Urban | Urban | Urban | Urban | Urban |
| School emphasis | Media, Sports | Music, Sports | Music, Sports | none | Languages, Science and research |
| Size of school (grade 10–12) | ≈500 | ≈550 | ≈650 | ≈850 | ≈450 |
| Number of participants (Male %) | 322 (32%) | 229 (34%) | 353 (40%) | 521 (40%) | 279 (23%) |
| Average grade of participant Mean (SD) | 8.97 (0.47) | 8.88 (0.17) | 8.70 (0.60) | 8.85 (0.49) | 9.67 (0.24) |
| Median grade of participants | 8.94 | 8.88 | 8.65 | 8.82 | 9.71 |
| Lowest grade of participant * | 7.76 | 7.47 | 7.41 | 7.24 | 9.00 |

\* In Finland, students are given a grade between 4–10, where 4 = fail, 8 = good.

**Table 2.** Measurements used in this study.

| What Was Measured? | Number of Items | Scale | Further Information |
|---|---|---|---|
| Climate change knowledge | 10 | multiple choice | Original questionnaire by Libarkin et al. [37] contains 21 items. We chose 10 items based on Rasch analysis from our previous study [14] concerning levels of difficulties and overall response time of the questionnaire. |
| Value | | | |
| Biospheric | 4 | −1–7 | This questionnaire, developed by Steg et al. [21] examines individuals biospheric, altruistic, egoistic and hedonic values. |
| Altruistic | 4 | | |
| Hedonic | 4 | | |
| Egoistic | 3 | | |
| Willingness for mitigative action | | | The questionnaire measures student's willingness to take climate action in three domains: as individuals, as members of a group and as future citizens (e.g., through career choices). This is a new questionnaire, inspired by the findings of Vesterinen et al. [38]. |
| Individual action | 4 | 1 to 5 | |
| Group action | 4 | | |
| Emotion | 3 | 1 to 5 | Three questions examined students' concern and emotions towards climate change. |
| School support: | | | This questionnaire examines how students perceive their schools to support them in agency and taking up future careers related to climate change. This is a new questionnaire and is inspired by the relevance framework (see [39]). |
| Student agency | 4 | 1 to 5 | |
| Future career | 4 | | |
| Supportive teacher | 4 | 1 to 5 | This questionnaire, developed by Ojala [40] examines how students perceive their teachers to talk about climate change. This study used three of the questions that measure teachers' positive outlook. |
| Responsibility | 3 | 1 to 10 | This questionnaire measures who individuals consider responsible for climate change mitigation. This is a new questionnaire developed for this study. |

### 2.3. Data Analysis

Initial data analysis was conducted to determine how the data should be grouped. To examine whether there is a difference between the average-ability students (Group 1) and the gifted students (Group 2, i.e., the top 10% in academic achievement) attending the normal schools, a t-test was conducted. The results showed that gifted students in normal schools (M = 6.38, SD = 1.42) had a significantly higher level of knowledge of climate change issues (t = −7.10, $df$ = 191.41, $p < 0.001$, d = 0.56) than average-ability students (M = 5.48, SD = 1.78). On the other hand, the average scores on climate knowledge did not show statistically significant differences between Group 2 and Group 3 (the gifted students at a gifted school, M = 6.70, SD = 1.35). Therefore, Group 1 and Group 2 were considered distinct from each other while Group 2 and Group 3 indicated a similarity.

Second, we assessed the validity and reliability of the constructs using exploratory factor analysis (EFA), confirmatory factor analysis (CFA), and Cronbach's alpha values. Initially, we randomly divided our sample into two equal parts and conducted EFA on one half, while the other half was used for CFA. For EFA, we applied the principal axis factoring

with varimax rotation, and we considered factor loadings higher than 0.5 to belong to the respective factors. Subsequently, we performed CFA to confirm the factors identified by EFA, incorporating all latent variables under their specific factors based on the EFA results. The model fit indices indicated a satisfactory fit (CFI = 0.93, TLI = 0.92, RMSEA = 0.04). However, due to a low factor loading (0.44) for item SUP1, as presented in Table 3, we excluded it from further analyses. Finally, each factor exhibited a Cronbach's alpha value higher than 0.7, indicating good reliability of the constructs.

**Table 3.** Reliability and validity results.

| Category | Subcategory | Item | EFA | CFA | Cronbach |
|---|---|---|---|---|---|
| Value | Biospheric | BIO1 | 0.77 | 0.69 | 0.86 |
| | | BIO2 | 0.84 | 0.69 | |
| | | BIO3 | 0.80 | 0.83 | |
| | | BIO4 | 0.67 | 0.83 | |
| | Altruistic | ALT1 | 0.71 | 0.62 | 0.72 |
| | | ALT2 | 0.61 | 0.59 | |
| | | ALT3 | 0.77 | 0.76 | |
| | | ALT4 | 0.72 | 0.59 | |
| | Hedonic | HED1 | 0.82 | 0.75 | 0.75 |
| | | HED2 | 0.87 | 0.88 | |
| | | HED3 | 0.73 | 0.55 | |
| | | HED4 | 0.51 | 0.53 | |
| | Egoistic | EGO1 | 0.80 | 0.72 | 0.79 |
| | | EGO2 | 0.79 | 0.71 | |
| | | EGO3 | 0.86 | 0.85 | |
| Willingness for mitigative action | Individual action | I-ACT1 | 0.81 | 0.75 | 0.84 |
| | | I-ACT2 | 0.80 | 0.78 | |
| | | I-ACT3 | 0.84 | 0.80 | |
| | | I-ACT4 | 0.68 | 0.71 | |
| | Group action | G-ACT1 | 0.78 | 0.71 | 0.85 |
| | | G-ACT2 | 0.82 | 0.81 | |
| | | G-ACT3 | 0.85 | 0.81 | |
| | | G-ACT4 | 0.70 | 0.72 | |
| Emotion | Environmental concern | EMO1 | 0.88 | 0.84 | 0.86 |
| | | EMO2 | 0.91 | 0.87 | |
| | | EMO3 | 0.85 | 0.73 | |
| School support | Student agency | AGE1 | 0.71 | 0.70 | 0.78 |
| | | AGE2 | 0.75 | 0.71 | |
| | | AGE3 | 0.71 | 0.71 | |
| | | AGE4 | 0.71 | 0.63 | |
| | Future career | FUT1 | 0.82 | 0.79 | 0.90 |
| | | FUT2 | 0.88 | 0.86 | |
| | | FUT3 | 0.86 | 0.88 | |
| | | FUT4 | 0.83 | 0.82 | |
| | Supportive teacher | SUP1 * | 0.79 | 0.44 | 0.77 |
| | | SUP2 | 0.70 | 0.70 | |
| | | SUP3 | 0.71 | 0.84 | |
| | | SUP4 | 0.70 | 0.82 | |

* SUP1 was removed for further analyses due to the low factor loading (0.44 in CFA).

Third, after confirming validity and reliability, we conducted measurement invariance tests before latent mean analyses. Specifically, the model's configural, metric, and scalar invariances were assessed and compared across groups. The configural invariance model assumes the same number of factors and items across groups without imposing equality constraints on other parameters. The results of the configural invariance measurement indicates that the variables being studied measure the same constructs across groups. Following that, metric invariance is evaluated by constraining factor loadings across groups. If the results demonstrate factor loading invariance, it suggests that the measures are operating on the same scale. Lastly, scalar invariance is tested by constraining both factor loadings and item intercepts across groups. If no significant differences are observed, latent means can be compared across groups. For these model comparisons, two indices, ΔCFI and ΔTLI, were assessed. To confirm invariance between the models, ΔCFI should be equal

to or less than 0.01, and ΔTLI should be equal to or less than 0.05 [41]. According to the result, no differences were found between configural, metric, and scalar models for the motivation factors (ΔCFI < 0.01, ΔTLI < 0.05) as presented in Table 4.

**Table 4.** Measurement invariance results.

| Model | $\chi^2$ (*df*) | RMSEA | CFI | ΔCFI | TLI | ΔTLI |
|---|---|---|---|---|---|---|
| 1 Configural | 3831.23 (1860) | 0.043 | 0.928 | | 0.919 | |
| 2 Metric | 3902.18 (1916) | 0.043 | 0.928 | 0.000 | 0.921 | 0.002 |
| 3 Scalar | 4045.75 (1972) | 0.043 | 0.925 | 0.003 | 0.919 | 0.002 |

Finally, we compared latent means between the groups to answer our research questions and the results are presented in the following section. For all these structural equation modeling analyses, Mplus 7.4 was used with the maximum likelihood with robust standard errors and the Chi-squared (MLR) estimator and missing data were estimated using full information maximum likelihood estimation (FIML) [42]. Traditional cutoff values were applied for assessing the quality of measurement and structural model fit ([43] the root-mean-square error of approximation (RMSEA) was below 0.05 or 0.08, the comparative fit index (CFI) and Tucker–Lewis index (TLI) were above 0.90 or 0.95).

## 3. Results

*RQ 1: How do academically gifted students' climate competencies differ from other students' competencies?*

First, we compared latent mean differences between the three groups while controlling gender and climate change knowledge. Concerning the value scales, as shown in Table 5, the average-ability students in the normal schools (hereafter Group 1) indicated higher hedonic and egoistic values than gifted students from the gifted school (hereafter Group 3). Except the value scales, on the other hand, Group 3 students indicated higher latent means than Group 1 students in all other measured constructs such as willingness, concern, and school environment. In other words, Group 3 students had more environmental concerns, they were willing to take more climate action in different domains, they viewed their schools' climate change education more positively and they even had lower non-environmentally friendly values (hedonic and egoistic values) comparing to Group 1.

**Table 5.** Mean, standard deviation, and latent mean differences between three groups.

| Category | Subcategory | Group 1 N = 1281 | Group 2 N = 144 | Group 3 N = 278 | G2 vs. G1 LMD | *d* | G3 vs. G1 LMD | *d* | G3 vs. G2 LMD | *d* |
|---|---|---|---|---|---|---|---|---|---|---|
| Value | Biospheric | 6.42 (1.29) | 6.32 (1.02) | 6.52 (1.10) | −0.07 | 0.09 | 0.12 | 0.08 | 0.19 | 0.19 |
| | Altruistic | 7.08 (1.07) | 7.06 (0.84) | 7.09 (0.99) | −0.01 | 0.02 | 0.02 | 0.01 | 0.03 | 0.03 |
| | Hedonic | 4.41 (1.32) | 4.08 (1.14) | 4.13 (1.29) | −0.32 * | 0.27 | −0.29 * | 0.21 | 0.04 | 0.04 |
| | Egoistic | 6.86 (1.20) | 6.76 (1.08) | 6.53 (1.10) | −0.11 | 0.09 | −0.31 ** | 0.29 | −0.20 | 0.21 |
| Willingness for mitigative action | Individual action | 3.41 (0.76) | 3.72 (0.67) | 3.81 (0.73) | 0.28 ** | 0.43 | 0.35 ** | 0.54 | 0.08 | 0.13 |
| | Group action | 2.23 (0.90) | 2.46 (0.81) | 2.68 (0.9) | 0.20 * | 0.27 | 0.43 ** | 0.50 | 0.23 * | 0.26 |
| Emotion | Environmental concern | 2.64 (0.94) | 3.02 (0.81) | 3.15 (0.96) | 0.40 ** | 0.43 | 0.48 ** | 0.54 | 0.08 | 0.15 |
| | Student agency | 2.31 (0.59) | 2.29 (0.52) | 2.56 (0.64) | −0.01 | 0.04 | 0.26 ** | 0.41 | 0.27 ** | 0.46 |
| School support | Future career | 1.90 (0.69) | 1.90 (0.65) | 2.17 (0.74) | 0.01 | 0.00 | 0.26 ** | 0.38 | 0.25 ** | 0.39 |
| | Supportive teacher | 2.29 (0.69) | 2.18 (0.63) | 2.58 (0.75) | −0.06 | 0.17 | 0.19 ** | 0.40 | 0.25 ** | 0.58 |

Note. * *p* < 0.01, ** *p* < 0.001, Group 1 (G1): average-ability students at normal schools, Group 2 (G2): gifted students at normal schools, Group 3 (G3): gifted students at gifted schools, LMD: Latent Mean Difference.

However, when comparing gifted students from normal schools (hereafter Group 2) to Group 1, the differences between the groups become less distinct. Namely, Group 2 students showed a higher willingness to take individual and group action as well as environmental concerns but did not show differences in their perceptions of the school environment. Interestingly, we also found a difference between the gifted student groups regarding the perceived school environment. That is, the gifted students in the gifted school (Group 3) showed better perceptions of school environments concerning climate change

education than the gifted students in the normal schools (Group 2). Additionally, Group 3 students presented a higher willingness for group action than Group 2 students.

Finally, we also found some differences in how gifted and average-ability students view responsibility. All groups viewed politicians as most responsible, businesses and individuals as least responsible (see Table 6). However, gifted students (both Group 2 and 3) viewed the responsibility of all three entities as higher than average-ability students, though statistically significant differences were only seen in two of the groups.

**Table 6.** Mean, standard deviation, and observed mean differences between three groups (responsibility items).

| Category | Subcategory | Group 1 | Group 2 | Group 3 | G2 vs. G1 $d$ | G 3 vs. G1 $d$ | G3 vs. G2 $d$ | F |
|---|---|---|---|---|---|---|---|---|
| | Individual | 6.31 (2.11) | 6.63 (1.93) | 6.58 (2.06) | 0.16 | 0.13 | 0.03 | F = 3.02 |
| Responsibility | Politicians | 7.96 (1.73) [a]* | 8.41 (1.49) [b]** | 8.71 (1.25) [a]*[b]** | 0.28 | 0.50 | 0.22 | F = 27.23 ** |
| | Business company | 7.76 (1.89) [a]* | 8.14 (1.44) [b]** | 8.58 (1.54) [a]*[b]** | 0.23 | 0.48 | 0.30 | F = 25.64 ** |

Note. * $p < 0.01$, ** $p < 0.001$, [a] Significant difference between G1 and G3. [b] Significant difference between G2 and G3. Group 1 (G1): average-ability students at normal schools, Group 2 (G2): gifted students at normal schools, Group 3 (G3): gifted students at gifted schools

*RQ 2: What type of effect does a school have on academically gifted students' climate competencies?*

As the results above showed that there are differences between gifted students from normal schools (Group 2) and gifted students from a gifted school (Group 3), we explored these differences in more detail. Again, comparing the two groups of gifted students in Table 5, we found they did not have differences in their knowledge, values, or individual action. However, gifted students from the gifted schools (Group 3) were more willing to take societal climate actions, and they perceived their education to provide them with more relevant skills for their everyday lives and their future careers. Furthermore, the students perceived their teachers talked about climate change in a more relevant way. Accordingly, we could assume that while factors that are more relevant to individual dimensions such as knowledge and values were more influential on individual actions, factors that are more relevant to school dimensions such as having supportive teachers or school environments equipping students for future careers were more effectful on willingness in group actions. Interestingly, the differences between Group 2 and Groups 3 were not individual dimensions but school dimensions. Thus, we investigated the relationships between the school dimension factors as shown in Figure 1 to understand the reasons as to why students in the gifted school (Group 3) perceive they get more from their education. To be specific, we wanted to know whether this difference between the groups was (i) because of differences in what actually happens in classrooms or (ii) because of what possibly happens in the school hallways when like-minded, gifted students come together. For this, we created a dummy variable (0 = Group 2, 1 = Group 3) measuring the school effect and controlled gender and knowledge effects.

According to the results as presented in Figure 1, the school effect (dummy) variable indicates significant positive correlation with all school environments factors (0.49, 0.61, and 0.40 with Student agency, Supportive teacher, and Future career, respectively). That is, similar to the results presented in Table 5, when gifted students were placed in the gifted school, they were more likely to value their school's climate education compared to the gifted students at normal schools. At the same time, the school effect variable indicates a direct effect (B = 0.23, $p < 0.05$) on willingness to environmental group actions after controlling for the effects of the three school factors on the group actions. Thus, it seems likely that the differences between Group 3 and Group 2 cannot be merely explained by differences in teacher competencies or school climate education, but rather, by some other factors such as what happens in the hallways of schools where gifted students come together.

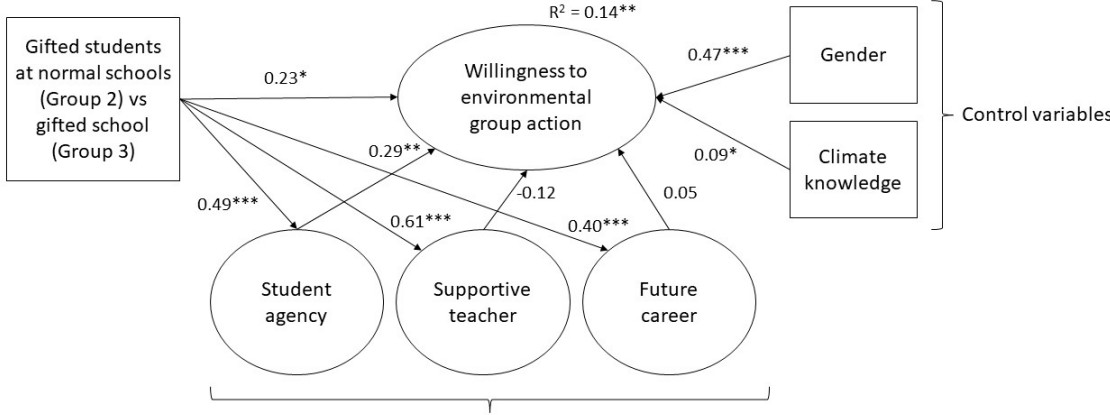

**Figure 1.** Effect of different schooling (normal vs. gifted) on gifted students' willingness. Note. RMSEA = 0.044, CFI = 0.959, TLI = 0.951. The values presented are standardized coefficients. Ellipse: latent variable. Rectangle: observed variable. * $p < 0.05$, ** $p < 0.01$, *** $p < 0.001$.

## 4. Discussion

In recent years there has been much discussion on developing students' competencies to take meaningful climate action. However, though previous studies acknowledge that academically gifted students may be more concerned about environmental issues and are morally sensitive when it comes to environmental issues [32], the relationship between academic giftedness and willingness to take climate actions has received little attention. This study contributes to this discussion through two main findings:

1. Academically gifted students show more climate competencies, including more willingness to take climate actions, than average-ability students and
2. Academically gifted students that attend schools for gifted students show more willingness to take societal actions related to climate change than gifted students from normal schools.

These findings have major implications to the research field, as we discuss below.

The findings of this study suggest that gifted students are more likely to have action competence than their peers. This notion is supported by the finding that academically gifted students show more knowledge, concern and willingness to take climate action than their peers. Furthermore, the notion is supported by previous studies, which have discussed how gifted students tend to show a high sense of responsibility, good problem-solving skills and they enjoy tackling big challenges (see, e.g., [30,32,34]), all of which are qualities of action competent citizens. Similarly, these are also important components of transformational giftedness. Referring to the definition of transformational giftedness, it can be hard to define what counts as an "extraordinary and meaningful contribution" to environmental issues, but our results indicate that gifted and talented students score high on constructs related to willingness to undertake societal actions, such as participating in public demonstrations and decision-making processes, as well as challenging politicians and businesses to do more to mitigate climate change. Researchers and stakeholders tend to agree that such actions are impactful and potentially transformative for society [1], supporting the notion that academically gifted students have more readiness to become transformationally gifted than their peers.

Furthermore, the core values of academically gifted students show some signs of higher readiness for transformational giftedness than their peers. Namely, academically gifted students showed lower hedonic values than their peers. This finding makes sense, as hedonic values coincide with seeking pleasure and instant gratification, something that may not be a good recipe for academic success. As hedonic values also have a negative correlation with pro-environmental behavior [21,22], values may also have an indirect effect on why academically gifted students showed more willingness to take pro-environmental

actions than their peers. That said, we need to be careful about how much we can read into this result as students in all three groups considered other values to be more important to them than hedonic values. According to the Value-Belief-Norms theory [44] an individuals' core values influence their actions. As in our study other values were more dominant than hedonic values, it is uncertain whether these low levels of hedonic values would have a significant impact on how the different groups act, despite finding a statistically significant difference in hedonic values. In other words, it is possible that the core values overrule the hedonic values in all three groups just as strongly. Further research should be conducted to examine whether differences in low-priority values truly have an impact on an individual's life, or whether more important values "override" such low-priority values.

### 4.1. Differences in Schools

One of the aims of education should be to train students to become active, transformationally gifted citizens. Interestingly, gifted students in the gifted school showed higher willingness to take societal actions than their gifted peers attending normal schools. This is despite the fact that the gifted students in both groups did not show differences in the level of their climate change knowledge. A supportive school environment seems to play a key role in developing readiness towards societal action, as the students in the gifted school perceived their teachers to be more supportive, and their climate education to be more relevant for them. As this study did not examine how climate change education was implemented in the schools, we don't know exactly why the students in the gifted school perceived their education to be more relevant. There are at least two, partially contradicting points of view. The first way to look at it is to assume that the quality of the education is better in the gifted schools, because a prestigious school may attract more competent teachers. However, even if the teachers were more competent in teaching their given subjects (though we don't know this), there is little reason to believe that this subject-specific confidence would translate into them teaching more about climate change per se. Furthermore, even if there were a difference in teacher competence, it is not translated into students having higher levels of knowledge on climate change than their peers, as seen in the results. Therefore, the more plausible explanation is that it may be so that climate change education in the gifted and normal school are more or less similar, but the students in the gifted school *perceive* their education to be more relevant for one reason or another. For instance, it is possible that the students in the gifted school merely perceive their education to be more relevant, due to psychological biases, such as the halo effect or endowment effect [45]. Afterall, they are attending a prestigious school, to which it is hard to get into, so one might assume that the quality of teaching in that school must also be better. It may not make a difference whether this subjective view is true or not, as studies on the halo effect have shown that one's perceptions affect how much that thing is cherished. In the case that education is perceived as relevant, this may result in a higher level of engagement and therefore, better learning results. Another option is that in the gifted school students are surrounded by other gifted students, impacting the type of discussions students engage in, not only during class, but also, outside of class. These "hallway discussion" may impact how relevant students see their school experience, which may then be projected into how relevant students see classroom education. In fact, based on our findings it seems that the group differences between the two gifted groups are not merely explained by differences in teacher competencies or the relevance of what happens in classrooms. Rather, some other factors such as socialization seem to be at play when gifted students come together. As students in the school for gifted are more willing to take societal actions than other gifted students, it could be that in the gifted school different social norms have formed. This would be in line with Ajzen's Theory of Planned Behavior [25], which describes social norms as an important component of pro-environmental behavior.

### 4.2. Limitations of the Study

It was beyond the scope of this study to examine the differences in school curricula or teachers' teaching methods. As mentioned, the school for gifted students had an emphasis on science and languages, meaning that students in this school had the *opportunity* to take more

courses in these subjects. In this study we did not examine which specific courses students had taken. However, we can assume that the school emphasis in itself does not have a major contribution to our results for two reasons. First, according to the national curriculum [8], climate change issues are mainly addressed in science courses, which are compulsory to all students in all schools. Furthermore, based on the names of these extra courses, provided by the school for gifted students, there is no indication that the courses examine climate change or sustainability issues. Rather, they mainly include courses such as lab courses, astronomy, and review-courses. Second, climate change issues are addressed not only in science, but also in other school subjects, such as in ethics. Therefore, having a science-focused school does not ensure that climate change issues will be dealt with more in such schools. In fact, as the Finnish curriculum is open ended, the teachers in Finland are given a lot of autonomy in how they interpret and implement the curriculum. Therefore, the interests of individual teachers tend to have a bigger impact on CC-Ed than the curriculum or a specific school emphasis (see, e.g., [9]). That said, an in-depth analysis, including teacher interviews, would have been beneficial to explore school differences and differences in teaching practices. As this was beyond the scope of this study, the results need to be examined with caution, as we were only able to examine a few of the various cofounding factors at play (e.g., type of school, focus, classroom discussion, teacher competence, teacher interests towards CC, peer relationships, family influence etc.)

### 4.3. Supporting Gifted Students

To become action competent or transformationally gifted, education must go beyond teaching about the science of climate change. In all, the participants in this study had fairly good knowledge on climate change, though we did see differences between gifted and average-ability students. However, in our study we saw differences among the students even when we controlled for gender and knowledge. This suggests that climate competencies, especially willingness to take climate action, cannot be explained merely by gender differences or differences in knowledge. Rather, other aspects, such as moral sensitivity and sense of responsibility may be at play.

We know from previous studies (e.g., [13]), that students need to be provided with opportunities to work with authentic, real-world dilemmas and problems. Authentic learning can take place when the challenges in learning are situated in some meaningful real-world tasks, solving real-world problems. Moreover, schools need to help students develop the skills to collaborate, and work in teams. This is especially important when dealing with multidisciplinary issues, such as climate change. As teamwork requires ethical and moral sensitivity in order to understand the other members' views, attitudes and values, this may be easier for academically gifted students, as they tend to rank higher in moral reasoning and ethical sensitivity than their average-ability peers [29–31].

Furthermore, to support gifted students it is important that their learning goals are ambitious enough [46] and are aligned with their abilities. In the case of gifted students, it is important that they have a chance to create something new and are guided to reflect the purposes of their learning with the beyond-the-self orientation, supporting the development of transformational giftedness. In other words, the learning goals should be meaningful to the students, while contributing beyond the self to make the world a better place. In the learning process it is important to receive feedback from the learning results. The Authors argue that gifted students need to learn to appreciate the importance of both receiving and giving peer-review in constructive and ethical ways [46]. Additionally, the learning results should not only be assessed with the criteria of excellence, but also with ethics. By also assessing how a school project enhances the wellbeing of humankind, and not only some gifted individuals, but the evaluation can also promote transformational giftedness.

As a concrete example, such a learning approach has been implemented in a non-formal education program for gifted students, where the students were given real-world problems by industry leading companies and universities to solve (see [47]). Over the period of the projects, students not only increased their knowledge and developed creative solutions to real-world problems, but the projects also opened academic and professional opportunities for the

students. Furthermore, it helped gifted students get to know each other better, meet experts and have fun together, all while having engaging and deep conversations on socio-scientific and environmental issues. Though this example is from a non-formal education setting, many of the same principles can be applied to formal education. However, it may be easier to implement such learning approaches in gifted schools, as all the academically gifted students already have a good level of base knowledge needed to work with real-world problems. Furthermore, they show high levels of engagement and interest towards working with global issues [32,47]. As gifted students in normal schools seem less engaged than those in gifted schools, our results indicate that they need more support in becoming transformationally gifted. To do so, teachers first need to recognize gifted students, and then provide them with engaging and challenging enough tasks, as discussed above. At times, gifted students should also be connected with other gifted students, to challenge and inspire each other. By taking an active role in supporting the gifted students, the teacher can help them become transformationally gifted, helping solve the local and global problems of today and tomorrow.

**Author Contributions:** Conceptualization, S.T.; data curation, S.T. and J.K.; formal analysis, J.K.; methodology, J.K. and S.T.; investigation, S.T.; validation, J.K. and S.T.; formal analysis, J.K; writing—original draft preparation, review and editing S.T., J.K. and K.T.; visualization, J.K. and S.T.; project administration, S.T.; funding acquisition, S.T. and K.T. All authors have read and agreed to the published version of the manuscript.

**Funding:** This research received no external funding.

**Institutional Review Board Statement:** Ethical review and approval were waived, according to FABRI (2012) guidelines. More specifically, ethical reviews were not needed as participants were over 15 years old, the questions were not sensitive in nature, and the participants had the right to refuse to participate in the study.

**Informed Consent Statement:** Informed consent was obtained from all subjects involved in the study.

**Data Availability Statement:** The data presented in this study are available on request from the corresponding author. The data are not publicly available as further data collection and analysis is still ongoing. Data-rights may also restrict the availability of data.

**Conflicts of Interest:** The authors declare no conflict of interest.

## Appendix A

**Table A1.** Survey questions included in this study.

| Category | Items |
|---|---|
| Knowledge (see Libarkin et al., 2018) [37] | Choose the right answer (multiple choice): <br> • According to climate scientists, how has the amount of carbon dioxide in the atmosphere changed since the start of the Industrial Revolution 150 years ago? <br> • According to climate scientists, which of the following statements about global warming over the past 50 years is most accurate? <br> • Which is the best description of the differences between climate and weather? <br> • Which of the following contributes to the transfer of thermal energy from place to place around the Earth? <br> • How does sunlight affect temperature on Earth? <br> • Which of the following will occur if the amount of ice floating in the ocean decreases? <br> • Which of the following would most likely occur if the oceans stopped absorbing carbon dioxide? <br> • Which is the best definition of a positive feedback loop in the climate system? <br> • Which of the following is the best definition of a greenhouse gas? <br> • How much incoming sunlight do greenhouse gases absorb? |

**Table A1.** *Cont.*

| Category | Items |
|---|---|
| Value (see Steg et al., 2014) [21] | Answer the following questions using the following scale (−1–7). Give the highest score (6 or 7) only to one or two of the principles which are most important to you. <br>• EQUALITY: Equal opportunity for all <br>• RESPECTING THE EARTH: harmony with other species <br>• SOCIAL POWER: control over others, dominance <br>• PLEASURE: joy, gratification of desires <br>• UNITY WITH NATURE: fitting into nature <br>• A WORLD AT PEACE: free of war and conflict <br>• WEALTH: material possessions, money <br>• AUTHORITY: the right to lead or command <br>• SOCIAL JUSTICE: correcting injustice, care for the weak <br>• ENJOYING LIFE: enjoying food, sports, leisure, etc. <br>• PROTECTING THE ENVIRONMENT: preserving nature <br>• INFLUENTIAL: having an impact on people and events <br>• HELPFUL: working for the welfare of others <br>• PREVENTING POLLUTION: protecting natural resources <br>• SELF-INDULGENT: doing pleasant things |
| Willingness to take individual action | How much effort are you willing to put into each of the following activities? <br>• Making lifestyle choices that will have a minimal negative impact on climate change. <br>• Finding out which products and services cause minimal harm to the climate. <br>• Reducing carbon emissions in my daily life <br>• Talking to friends and family about climate change related issues so that we can all become more aware of what to do about the problem. |
| Willingness to take group action | How much effort are you willing to put into each of the following activities? <br>• Challenging politicians and businesses to do more to mitigate climate change <br>• Be a member of a local or national youth group/forum that promotes climate issues. <br>• Seek opportunities to participate in decision-making processes at national and international levels regarding climate issues. <br>• Participate in public demonstrations (e.g., climate strikes) to support the climate change movement. |
| Environmental concern | On a scale of 1–5 <br>• How worried are you about climate change? <br>• How anxious are you about climate change? <br>• How much guilt do you feel about climate change? |
| School support: Student agency | Use the following scale to answers the questions: (1 = does not apply at all; 4 = applies very well) <br>• School teaching and activities have provided me with interesting new knowledge, skills and experiences about climate change related issues. <br>• School teaching and activities have given me ideas on how I can put knowledge, skills and experiences about climate change into practice in my everyday life. <br>• School teaching and activities have enabled me to understand how I can help my local community and my country to mitigate climate change. <br>• School teaching and activities have enabled me to understand my own role as a member of society in mitigating climate change |

**Table A1.** *Cont.*

| Category | Items |
|---|---|
| School support: Future career | Use the following scale (1–4) to answers the questions: (1 = does not apply at all; 4 = applies very well) <br> • School teaching and activities have enabled me to get ideas on what type of career I could pursue in order to work with climate change related issues. <br> • School teaching and activities have helped me understand what type of further education is required of me if I wish to pursue a career where I could work with climate change related issues. <br> • School teaching and activities have enabled me to understand the skills that are necessary in the professions related to climate change. <br> • School teaching and activities have helped to understand what it could be like to work in a career position related to climate change |
| School support: Supportive teacher (see Ojala, 2015) [40] | Use the following scale (1–4) to answers the questions: (1 = does not apply at all; 4 = applies very well) <br> • I have teachers who talk about societal and environmental issues related to climate change in a thought-provoking way. <br> • I have teachers who take up how I, as a young person, can alleviate various societal and environmental problems related to climate change. <br> • I have teachers who in talking about societal and environmental problems related to climate change indicate possible ways to solve those problems in the future. |

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
