# Peer review of "Climate Competencies of Finnish Gifted and Average-Ability High School Students"

_education, doi:10.3390/educsci13080840_

Round 1

Reviewer 1 Report

Review of „Climate competencies of Finnish gifted and average-ability high school students“

The authors of this submission aim at investigating differences between academically gifted and non-gifted students in terms of climate competencies. There are several major points that must be addressed before this ms. can be reassessed for potential publication, as I describe below:

-          My biggest concern is that the used design of the present research appears to be tautological in nature. The authors describe that according to the curriculum in Finland, “climate issues” are actively taught in every school and at least implicitly in every subject starting from secondary school (p.2 second para). In their design, they assign academically gifted vs. non-gifted students to groups according to school grades. Therefore, it is unsurprising that those with higher grades show higher climate competencies, because they obviously do better in school where these things are explicitly taught.

-          I am surprised that throughout the ms. the authors appear to be reluctant to the idea that educational content/quality may differ between the school for the gifted students and those for the general student population (e.g., p.10). What would be the use of establishing schools for students with different needs, if there were no intentions to tailor the teaching content to those student populations? I suspect that teaching content will be denser and teaching speed will be higher in the school for the gifted. This would explain the differences between academically gifted students between different schools.

-          The paper is riddled with buzzwords and vague descriptions (e.g., “curriculum related barrier for transformational giftedness”, p.2). I would like to encourage the authors to provide concrete descriptions and clearly define the constructs that they refer to.

-          The data analysis section is vague. Why did you use a shortened climate change knowledge scale, in which way did you use Rasch (not “Ratch”, Table 2) analysis to select items, why did you re-examine the factor structure of the used scales (especially by means of EFA)? The scale labels of Table 2 are different from those in Table 3 (the item labels are useless). It is also unclear why only certain scales were reexamined in terms of factor structure (how was this determined?). The alpha-values in Table 3 suggest that there are actually multiple subscales within some of the measures from Table 2 that are nowhere mentioned. Item numbers from Table 2 do not always correspond to those in Table 3 (School support appears to have 12 items with 3 subscales according to Table 3, although there should be only 3 items according to Table 2).

-          The measurement invariance analysis approach should also be explained in more detail. Moreover, I suggest using the more common CFI-based criterion to evaluate measurement invariance between models instead of delta-TLI (a sign is missing for the first delta-TLI value in Table 4).

-          Test statistics are missing for most analyses (i.e., p/T/F values). Please add effect sizes for mean differences (e.g., Hedges g) to allow an evaluation of the meaningfulness of the respective effects.

-          It is unclear why the authors control for gender and climate change knowledge in their analyses. This is nowhere explained and in all likelihood not a good idea. Climate change knowledge is most certainly positively related to climate competencies. So why artificially deflate a salient effect by controlling for knowledge? I am also unsure what the reason for controlling for gender is.

-          The interpretation of the findings is questionable in several places. For instance, on p.10 (lines 335ff) the authors argue that one “[…] shouldn’t read too much into the differences in values, as hedonic values were ranked the lowest out of the four values among all three groups of students.”. The authors appear to assume that the absolute numerical value of a given scale entails information about the relevance of group differences. However, this is a misperception. The absolute value on a given scale is a function of the item difficulty (in this case, the probability of a given person with a certain characteristic to agree with the item), but not the relevance of observed group differences (i.e., if the respective scale is not referenced to a standardization sample). Therefore, the authors’ suggestion not to “read too much in the differences in values” is inappropriate.

The Quality of English is fine; some typos need to be weeded out.

Author Response

Thank you for reviewing our work. Please see the attachement for our response to your concerns, as well as the modified ms.

Reviewer 2 Report

I don't have many comments as I don't subscribe to your views of giftedness. 

Author Response

Thank you for reviewing our work.

Reviewer 3 Report

I have carefully reviewed your work and would like to provide feedback to help enhance the clarity, comprehensiveness, and overall quality of your paper. This is an improtant topic and I commend the authors for making an effort to provide empirical evidence towards the theory of Transformational Giftedness (TG).  Please find below some comments and suggestions that could help the readers clearly understand the goals and results of the paper.

Intro and lit review

For the definition and research questions, please clarify, the operational construct to be addressed by the research at times the researchers speak of climate competencies or transformational giftedness, or indicator of approaching transformational giftedness. Research questions relate to climate competencies which seems a proxy for transformational giftedness, I suggest briefly explain how the operationalization offered by the authors relates to Sternberg’s theory. Are these terms distinct or the same? Lines 134 and 139: Use consistent language, whether students have transformational giftedness (climate competence) or whether they approach TG seem to be different concepts. Similarly in Line 327 the authors speak of readiness for transformational giftedness. Line 320 transformational giftedness use concise throughout, not transformative.

Line 13.  use of language normal school vs general education school. Schools for formally identified gifted children?

Line 46. Coined the term.

Methods

Table 1 for clarity of the statistics, show Mean and SD for all measures. Provide information about student demographics. Further explanation of the sample composition is necessary. For example, in RQ 2 speaks of control for Gender, but no information is provided about the distribution of females, males across gifted/non-gifted and schools. 

Line 198 How was EFA and CFA conducted, on what sample, were those the same samples? Clarity and justification are needed as to why the researchers used structural equation modelling, rationales for model specification, the latent variables included and those excluded as well (since several latent variables were included in the first analysis). Rationale for the selected statistical controls.

Results

Line 191 report t-statistic, df, N, and p-value

Table 5, report comparison statistics with N, significance for all variables and include knowledge, as it seems that knowledge is one of the competencies evaluated here.

RQ2. Preferably report standardized coefficients the variables are in different scales. Why were values and emotion not included in the model? Were all measured items included in the latent variables?

Lines 292 through 295, explanation of the result seems editorialized and not related to current analysis. This could be addressed in the discussion, plausibly considering peer influence or other specific factors.

Discussion

Line 322 and 324, results indicate that gifted and talented students score high on constructs related to ….

Because of the schools had different foci, to what extent could climate competencies performance, as well as other measures, vary because of school focus (e.g., sports vs Science)?

Line 330-331 Sternberg considers that search for gratification in highly capable individuals leads to transactional giftedness which is not incompatible with academic success. Is there a way that your study can make a distinction between these two types of giftedness, and further argue with the evidence provided why the outcomes are related to transformational and not transactional. For example, consider the role of social desirability in answering the questionnaires, which Sternberg has previously addressed, arguing that the ultimate indicator of TG is action, not just knowledge or intention.

Line 363 Do the authors mean affect?

Lines 360 – 361 claims about halo effect in gifted children could be referenced/cited adding empirical evidence.

Lines 371 -376. Note that a plausible explanation could be related to the fact that the schools have different foci. Do students in the gifted school have more opportunities to learn, discuss, practice, climate competencies due to the focus on academics and science? How can curricular and extracurricular activities influence student outcomes when the focus is on sports or music vs language and science?

Line 379 Action competent?

Lines 379-428 While some of the discussion about how transformational giftedness can be promoted in schools and non-formal learning contexts, it seems that this discussion section is out of the scope of the study. This space in the paper could be use to address some of the limitations of the paper, including that relying on observational data might not provide direct evidence for curricular/instructional implications, especially when various confounding factors are at play (e.g. type of school, focus, classes and time spent in environmental/climate discussion/activities, peer relationships and influence, teacher quality, etc).

This last issue can be solved by adding a section on limitations/future directions and conclusions.

Author Response

Thank you for your  valuable feedback and review. Please see the attachement with our response to the issues raised.

Round 2

Reviewer 3 Report

Thank you for thoroughly addressing the comments.